# OFCOURSE: A Multi-Agent Reinforcement Learning Environment for Order Fulfillment

**Yiheng Zhu**[1], **Yang Zhan**[2,3],* **Xuankun Huang**[1], **Yuwei Chen**[1], **Yujie Chen**[1],
**Jiangwen Wei**[1], **Wei Feng**[1], **Yinzhi Zhou**[1], **Haoyuan Hu**[1], **Jieping Ye**[4]

[1]Department of Artificial Intelligence, Cainiao Network.
[2]Zhejiang University of Technology. [3]Hailiang Group Co., Ltd. [4]DAMO Academy.

`{zyh171911,huangxuankun.hxk,chenyuwei.chenyuwe,aisling.cyj,`
`jiangwen.wjw,fw222241,yinzhi.zyz,haoyuan.huhy}@cainiao.com,`
`zhanyang@zjut.edu.cn, yejieping.ye@alibaba-inc.com`

## Abstract

The dramatic growth of global e-commerce has led to a surge in demand for efficient and cost-effective order fulfillment which can increase customers' service levels and sellers' competitiveness. However, managing order fulfillment is challenging due to a series of interdependent online sequential decision-making problems. To clear this hurdle, rather than solving the problems separately as attempted in some recent researches, this paper proposes a method based on multi-agent reinforcement learning to integratively solve the series of interconnected problems, encompassing order handling, packing and pickup, storage, order consolidation, and last-mile delivery. In particular, we model the integrated problem as a Markov game, wherein a team of agents learns a joint policy via interacting with a simulated environment. Since no simulated environment supporting the complete order fulfillment problem exists, we devise Order Fulfillment COoperative mUlti-agent Reinforcement learning Scalable Environment (OFCOURSE) in the OpenAI Gym style, which allows reproduction and re-utilization to build customized applications. By constructing the fulfillment system in OFCOURSE, we optimize a joint policy that solves the integrated problem, facilitating sequential order-wise operations across all fulfillment units and minimizing the total cost of fulfilling all orders within the promised time. With OFCOURSE, we also demonstrate that the joint policy learned by multi-agent reinforcement learning outperforms the combination of locally optimal policies. The source code of OFCOURSE is available at: `https://github.com/GitYiheng/ofcourse`.

## 1 Introduction

The growth of e-commerce has increased the need for efficient and cost-effective Order Fulfillment (OF) services which are essential for upgrading customers' service levels and enhancing the sellers' competitiveness [Houde et al., 2017]. OF covers the entire process from order placement to order delivery, involving a series of online sequential decision-making problems across different fulfillment stages, including order handling, packing and pickup, storage, order consolidation, and last-mile delivery [Lin and Shaw, 1998, Croxton, 2003]. To reduce the overall cost of the complete OF problem, it is pivotal to approach it holistically, which poses two challenges. Firstly, order information is revealed progressively over time, while decisions must be made once the requests arrive without future information. Secondly, as OF involves managing numerous orders over multiple interconnected fulfillment units, all constituent decision-making stages are interdependent and require collective

---

*Yang Zhan (`zhanyang@zjut.edu.cn`) is the corresponding author.

consideration. Recent works have revealed that Multi-Agent Reinforcement Learning (MARL) [Zhang et al., 2021] is a well-established paradigm for handling these challenges with interacting constituents. However, due to the lack of a standard MARL framework for the complete OF problem, previous works often focused on solving individual OF subproblems, as reviewed in Section 2 and compared in Appendix C.

To enable the customization for various OF problems and facilitate the deployment of future MARL algorithms, this paper proposes a MARL framework for the complete OF problem. Specifically, we formulate the complete OF problem as a Markov Game (MG) where a team of fulfillment agents collectively learns to minimize the global fulfillment cost by interacting with an environment [Busoniu et al., 2008], as detailed in Section 3. To support this, we develop a simulated environment called OFCOURSE (Order Fulfillment COoperative mUlti-agent Reinforcement learning Scalable Environment). As described in Section 4, OFCOURSE conforms to the OpenAI Gym format [Brockman et al., 2016, Koul, 2019, Terry et al., 2021] and takes into account the heterogeneity, scalability, and variability in OF problems. In Section 5, we show how OFCOURSE can be used to customize OF problems in diverse contexts. Moreover, by conducting computational experiments, we demonstrate that the joint policy learned via MARL outperforms the combination of locally optimal policies.

## 2   Related Work

Order Fulfillment (OF) problems have been extensively studied in the fields of operations management [Liu et al., 2022a, Gopalakrishnan et al., 2023], supply chain management [Yang et al., 2021], operations research [Bayram and Cesaret, 2021, Jiang et al., 2022], and a recent one — machine learning. Here we focus on reviewing the stream that applies Reinforcement Learning (RL) techniques to solve OF problems, which are closely related to this paper.

A large number of works have proposed RL-based methods to solve decision-making problems in the complete OF. For example, to determine the processing sequences of orders (known as scheduling) in different processing units (warehouse pickers) in the **order handling** stage of OF, Zhang et al. [2020] used graph isomorphism networks for feature embedding and proximal policy optimization for policy learning to solve the problem in the static context. Liu et al. [2022b] solved the dynamic job shop scheduling problem using double deep Q-networks. To select a container (box) and determine the positions of items within the containers in the **packing and pickup** stage, Hu et al. [2017] employed a pointer network to learn the optimal sequence of items, and Duan et al. [2019] extended this approach by learning the orientations in a multi-task fashion. To ensure the appropriate inventory levels to meet the demand (known as the inventory management problem) in the **storage** stage of OF, Oroojlooyjadid et al. [2022] learned to optimize the solution using the deep Q-network. Later, De Moor et al. [2022] augmented this deep Q-network approach by reward shaping using existing well-performing heuristics to solve perishable inventory problems. To solve the well-known vehicle routing problem, which assigns orders to vehicles and determines the optimal route of each vehicle in the **last-mile delivery** stage of OF, Nazari et al. [2018] developed a policy network using a recurrent neural network decoder with the attention mechanism. James et al. [2019] proposed a model based on a graph convolutional network encoder and a recurrent neural network decoder to solve the online version of the vehicle routing problem, where the delivery requests were not static but arrived progressively. Additionally, Duan et al. [2020] constructed a model based on the graph convolutional network and trained the model using a hybrid learning method.

The above pioneering studies have demonstrated that RL is a promising approach to handle the complexity of OF problems. However, they only focused on addressing individual OF subproblems (refer to Appendix C for a comparison between precedent formulations and our contribution), which may yield suboptimal solutions when viewed from a holistic perspective. Additionally, they did not construct RL environments in standard formats, such as the OpenAI Gym style [Brockman et al., 2016, Koul, 2019, Terry et al., 2021], making it difficult to reproduce the results and reuse them to build customized applications.

Recently, some works have open-sourced standard RL format environments that covered a portion of the subproblems of OF. For example, Balaji et al. [2020] benchmarked online bin-packing, news vendor, and vehicle routing problems. Later, Hubbs et al. [2020] enriched this benchmark to incorporate knapsack, multi-period asset allocation, multi-echelon supply chain management, and

virtual machine assignment problems. However, they only supported single-agent RL, which cannot jointly handle multiple decision-making problems in OF. This paper contributes to the literature by filling the research gap of collectively solving the series of decision problems in OF and creating a standard OF environment in the OpenAI Gym format, which facilitates the deployment of future MARL algorithms in solving various OF problems.

## 3  Background

### 3.1  Order Fulfillment Problem

Order fulfillment involves delivering ordered items over a fulfillment system, with the aim to **minimize the overall fulfillment cost** within the promised time [Lin and Shaw, 1998, Croxton, 2003].

For an individual order, once a customer places it online, a record is created on the e-commerce platform server. Right after that, the order progresses to the order handling stage where it accumulates processing time and incurs costs related to server maintenance and customer service. Next, the order information arrives at the seller who packs the ordered items where a packing scheme is chosen from various options with different prices and processing time. Starting from there, the parcel is picked up by an agency, with multiple agencies available at various prices and time, and a decision is made on which to choose. Then, at the consolidation stage, the parcel can wait to be consolidated with others going to the same destination, taking longer but saving downstream costs. After that, the parcel undergoes warehouse storage, where a decision is made regarding whether to delay the delivery in order to free up logistics partner capacities at the next stage for other urgent parcels. At the last-mile delivery stage, a decision is made to select the most suitable logistics partner for delivery from multiple candidates offering different prices and delivery time. There are numerous such orders in a complete OF system.

In reality, OF systems vary significantly depending on the context, and their characteristics can be summarized as follows.

**(i) Heterogeneity:** Fulfillment stages are heterogeneous, as the aims and operations vary across different fulfillment stages. For example, the operations in the consolidation stage involve either waiting or consolidating existing orders, while the operations in the last-mile delivery stage correspond to different logistics partners to choose from. It is worth noting that even within the same stage, the fulfillment units from different fulfillment agents may not be identical. Taking the delivery stage for example, one fulfillment agent offers air transport as a candidate operation, while another fulfillment agent offers sea transport, resulting in significant differences in terms of cost and time.

**(ii) Scalability:** Fulfillment systems come in various sizes depending on the business contexts. For example, when it comes to domestic fulfillment processes that fulfill orders within a country, there are typically three fulfillment stages involved: local customers placing orders online, the e-commerce platform notifying the sellers, and the sellers delivering the ordered items directly to the customers' doorstep. On the other hand, a cross-border fulfillment process requires additional fulfillment stages such as the collection warehouse, distribution center, consolidation warehouse, line-haul warehouse, and more.

**(iii) Variability:** The fulfillment system is subject to change due to the constant variation of fulfillment resources. Fulfillment resources may become unavailable, as exemplified by cases where order processing servers go offline owing to policy or technical issues. Additionally, the set of available actions can also vary dynamically. For instance, last-mile delivery may be expanded to include an additional logistics partner after the onboarding of a new company.

To effectively model diverse OF tasks, a desirable MARL framework needs to account for the characteristics of heterogeneity, scalability, and variability. In Section 4, our proposed framework addresses these aspects, thereby enabling a comprehensive representation of diverse OF tasks.

Before elaborating on our approach, we first introduce key concepts and notations. In a fulfillment system, we define the fundamental building block as a **fulfillment unit**. For example, in Figure 1, the fulfillment unit in the order placement stage (the leftmost blue box) encapsulates the action of [order_creation] and observation of order information for the 2 illustrated orders. As illustrated in Figure 1, a fulfillment **agent**, $i \in \mathcal{I}$, connects fulfillment units into a fulfillment route, with each fulfillment unit realizing a fulfillment stage. Its **observation space** (the red box in Figure 1),

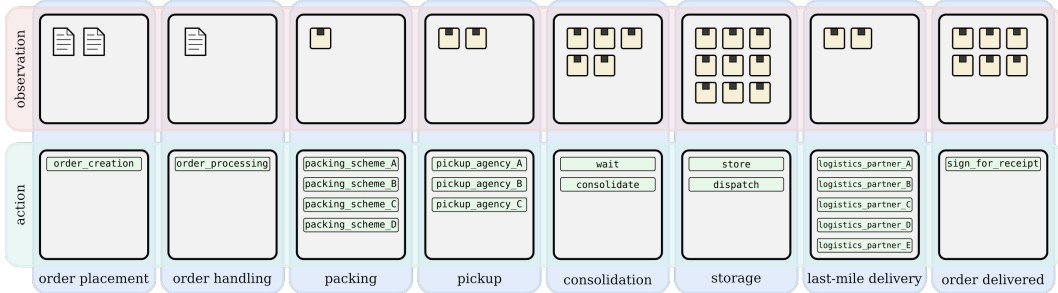

Figure 1: An example order fulfillment agent. The observation is highlighted by a red box and the action is emphasized by a green box. All fulfillment stages are marked by blue boxes.

$o^i$, combines order information across all of its fulfillment units. For example, this observation space can be represented by a feature matrix of size [num_unit, num_order, order_feat], where num_unit indicates the number of fulfillment units (8 fulfillment units in Figure 1), num_order signifies the current number of orders in the agent ([2, 1, 1, 2, 5, 9, 2, 6] in Figure 1) and order_feat denotes the feature dimension of each order. Its **action space** (the green box in Figure 1), $a^i$, concatenates the discrete actions across all of its fulfillment units. For example, in the pickup stage in Figure 1, the agent can select pickup_agency_B from the available operations of [pickup_agency_A, pickup_agency_B, pickup_agency_C]. Accordingly, at a decision step, the action of a fulfillment agent encompasses such immediate actions pertaining to all its carried orders (28 orders in Figure 1). A fulfillment system consists of $|\mathcal{I}| = L$ such fulfillment agents, where each fulfillment agent is responsible for making decisions for a specific portion of the orders. Here, orders compete for recourses across multiple fulfillment stages, so agents interact accordingly. In order to solve the complete OF problem by $L$ interacting agents, we need to find a joint **policy**, denoted as $\boldsymbol{\pi}(\boldsymbol{a}_t|\boldsymbol{o}_t) = \prod_{i=1}^{L} \pi^i(a_t^i|o_t^i)$ where $t$ is the decision step, that makes a series of decisions along the fulfillment system as described above in the MARL framework.

> **Problem Definition:** Order fulfillment involves transferring $|\mathcal{N}|$ orders from source to target fulfillment units in $T$ discrete steps. Order $j \in \mathcal{N}$, with promised fulfillment time $\tau_j$, is created at its source fulfillment unit at step $t_j^{init}$ and arrives at its target fulfillment unit at step $t_j^{trml}$. The total fulfillment cost of order $j$ is $c_j = \sum_{t=t_j^{init}}^{t_j^{trml}} c_{j,t} + p_j$ where $c_{j,t}$ is the step cost and $p_j$ is the penalty if $t_j^{trml} - t_j^{init} > \tau_j$. The objective of order fulfillment is to find the optimal (joint) policy to minimize the fulfillment cost of these $|\mathcal{N}|$ orders, $\boldsymbol{\pi} \leftarrow \operatorname{argmin}_{\boldsymbol{\pi}} \sum_{j=1}^{|\mathcal{N}|} c_j$.

### 3.2 Markov Game

The complete OF problem can be formulated as a Markov Game (MG) [Littman, 1994] based on the enumerated characteristics and notations in Section 3.1. In the literature, similar concepts of MG have been referred to as stochastic game [Shapley, 1953], multi-agent Markov decision processes [Boutilier, 1996, Lauer and Riedmiller, 2000], and Markov teams [Ho, 1980, Wang and Sandholm, 2002]. There has been a significant amount of research on a specific type of Markov game (MG) that involves partial observability in a cooperative setting, which is known as the decentralized partially observable Markov decision process (Dec-POMDP) [Bernstein et al., 2002, Oliehoek and Amato, 2016]. In this paper, to cover various OF scenarios, we formulate it as a general MG that is denoted by a tuple $\mathcal{M} = \langle \mathcal{I}, \mathcal{S}, \rho_0, \boldsymbol{\mathcal{A}}, P, R, \boldsymbol{\mathcal{O}}, \Omega, h, \gamma \rangle$, where $\mathcal{I}$ is a set of $L$ agents with a specific agent denoted by $i \in \mathcal{I}$, $\mathcal{S}$ signifies the state space, $\rho_0 \subset \mathcal{S}$ represents the initial state distribution, $\boldsymbol{\mathcal{A}} = \prod_{i=1}^{L} \mathcal{A}^i$ denotes the joint action space, $P : \mathcal{S} \times \boldsymbol{\mathcal{A}} \times \mathcal{S} \rightarrow [0, 1]$ is the transition function, $R : \mathcal{S} \times \boldsymbol{\mathcal{A}} \rightarrow \mathbb{R}^L$ represents the reward function, $\boldsymbol{\mathcal{O}} = \prod_{i=1}^{L} \mathcal{O}^i$ denotes the joint observation space, $\Omega : \mathcal{S} \times \boldsymbol{\mathcal{O}} \rightarrow [0, 1]$ is the observation function, $h \in \mathbb{N}^+$ is the time horizon, and $\gamma \in [0, 1)$ signifies the discount factor. For a given MG $\mathcal{M}$, the MARL objective is to find a joint policy $\boldsymbol{\pi} : \boldsymbol{\mathcal{O}} \times \boldsymbol{\mathcal{A}} \rightarrow [0, 1]$ to maximize the objective function $J(\boldsymbol{\pi}) = \mathbb{E}_{s_{0:h} \sim \rho_{\boldsymbol{\pi}}, \boldsymbol{a}_{1:h} \sim \boldsymbol{\pi}} \left[ \sum_{t=1}^{h} \gamma^t \boldsymbol{r}_t \right]$, where $\rho_{\boldsymbol{\pi}}(s) = \sum_{t=0}^{h} \gamma^t P(s_t = s)$ represents the discounted visitation frequencies, and $\boldsymbol{r}_t = R(s_t, \boldsymbol{a}_t|s_{t-1})$ denotes the joint reward

that is equivalent to the negative joint fulfillment cost (detailed calculation is provided in Section 4.5). To elaborate, an MG trajectory starts from initial state $s_0 \in \rho_0$. At step $t \in [1, h]$, facing state $s_{t-1} \in \mathcal{S}$, the team of agents jointly observe $\boldsymbol{o}_t = [o_t^1, \ldots, o_t^L] \in \mathcal{O}$ via $\Omega(\boldsymbol{o}_t | s_{t-1})$. Based on this joint observation $\boldsymbol{o}_t$, the team of agents samples a joint action $\boldsymbol{a}_t = [a_t^1, \ldots, a_t^L] \in \mathcal{A}$ according to the joint policy $\boldsymbol{\pi}(\boldsymbol{a}_t | \boldsymbol{o}_t) = \prod_{i=1}^{L} \pi^i(a_t^i | o_t^i)$. Simultaneously, the agent team receives a joint reward $\boldsymbol{r}_t = [r_t^1, \ldots, r_t^L] \in \mathbb{R}^L$ and transits to the next state $s_t$ with probability $P(s_t | s_{t-1}, \boldsymbol{a}_t)$.

# 4  OFCOURSE

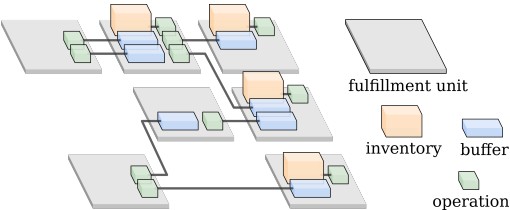

Figure 2: An example fulfillment system in OFCOURSE. A fulfillment unit (gray) is the basic building block, containing operations (green) and containers. Inventories (orange) and buffers (blue) are two types of containers. Orders are held in containers and transferred between containers via operations.

To address the complete OF problem by MARL, we are motivated to design an environment for the agent team to interact with. Considering the three characteristics described in Section 3.1, we devise Order Fulfillment COoperative mUlti-agent Reinforcement learning Scalable Environment (OFCOURSE) with three corresponding functionalities. **(i) Time-price metrics**: Different fulfillment units can have distinct properties. Accordingly, OFCOURSE focuses on recording the *price* and *time*, while neglecting other inconsistent details. Specifically, OFCOURSE maintains cumulative price and time for each order, updating them at every simulation step. **(ii) Modular design**: Fulfillment systems can vary significantly in scale. To accommodate this, OFCOURSE adopts a modular design approach, allowing the construction of fulfillment systems for different scenarios using basic building modules. **(iii) Step-wise update**: The status of fulfillment units is subject to constant change. To cope with this, OFCOURSE supports step-wise updates for module variables. The components of OFCOURSE will be described in detail in the remaining parts of this section.

## 4.1  Order

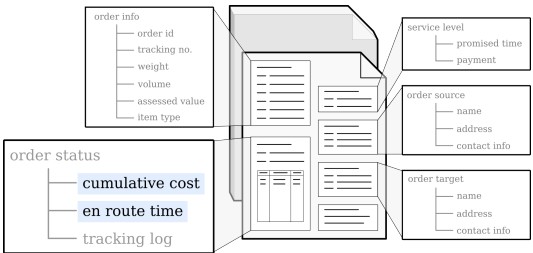

Figure 3: A typical order. The goal of order fulfillment is to minimize the overall cost of delivering all orders within promised time. Accordingly, OFCOURSE focuses on keeping track of the cumulative price and time, while neglecting other inconsistent details.

An order is the embodiment of a customer's demand, which can be either virtual or physical. For example, movies or music can be considered virtual orders, while cosmetics or electronics are physical orders. With the objective of minimizing the total cost of fulfilling all orders within the promised time, OFCOURSE keeps track of prices and time for all orders, as Figure 3 illustrates. As the time counter for an order increments at simulation step $t$, its step-wise financial cost is calculated by $c_t = c_t^{\text{ctr}} + c_t^{\text{op}}$, where $c_t^{\text{ctr}}$ denotes the container-related cost and $c_t^{\text{op}}$ represents the operation-associated cost. With regard to the container cost, an order incurs charges for being held by a specific container, which

depends on the container's quote based on its occupancy status. An example of this cost is the storage fee at a warehouse. As for the operation cost, an order incurs charges for being transferred from one container to another. For example, an e-commerce platform may charge for order processing and customer service.

## 4.2 Container

A container serves as the carrier of orders. Each container is capable of holding multiple orders, but each individual order can only be held by one container. For example, a server can act as a container for virtual orders such as movies, while a warehouse shelf can serve as a container for tangible orders like laptops.

**Resource:** In reality, the resources associated with containers are not unlimited. For example, the storage space in a warehouse can be exhausted. However, it is possible to rent additional storage space at a higher cost. Considering this aspect, we assign a specific resource to each container, where the same resource can be shared among multiple containers. For instance, orders traveling along different routes can share the same vehicle for a portion of their transportation. At simulation step $t$, the quote of a container depends on its resource usage, denoted as $c_t^{\text{ctr}} = g(u_t^{\text{ctr}})$, where $u_t^{\text{ctr}}$ represents the container's usage, and $g(\cdot)$ is a nonlinear function that calculates the quote based on the usage.

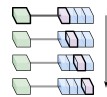

Figure 4: Buffer mechanism in OFCOURSE. A buffer consists of multiple chunks, where each chunk can hold multiple orders. At each simulation step, the orders in a chunk are transferred to its successor chunk.

**Buffer and Inventory:** There are two types of containers in OFCOURSE: buffers and inventories. Specifically, buffers carry geographically or procedurally dynamic orders, while inventories hold static ones. Conventionally, the estimated time of arrival (ETA) [Wang et al., 2018] is regarded as an order-wise feature. In OFCOURSE, the ETA of an order is indicated by its located chunk in a buffer. Figure 4 demonstrates the structure of a buffer, which consists of a sequence of chunks $(B_T, \ldots, B_{T+n}, B_O)$. At simulation step $T = t$, only the orders in chunk $B_t$ are operable. The orders in chunk $B_{t+n}$ reach $B_t$ after $n$ simulation steps. Meanwhile, outlier orders are temporarily stored in chunk $B_O$. Due to the heterogeneity of the fulfillment system, certain orders are processed in batches, some orders are handled sequentially, and others are handled individually. Using discrete buffers is a design compromise that ensures compatibility across heterogeneous fulfillment units. Additionally, using discrete buffers helps to preserve the morphological information of the fulfillment system. Another type of container is the inventory, which holds static orders. Orders stored in an inventory are subjected to recurring fees charged at each simulation step. For example, the storage fee in a warehouse is charged on a daily basis.

## 4.3 Operation

An operation is an abstract action of transferring orders between containers. For example, transporting an ordered item from a regional hub to a last-mile warehouse can be outsourced to different third-party logistics (3PL) companies. Consequently, the decision problem in OFCOURSE involves selecting a 3PL company from a pool of candidates, with each company represented as an individual operation.

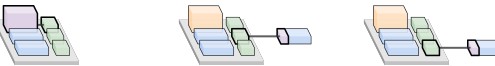

Figure 5: An example fulfillment unit with three operations in OFCOURSE. Activated operations are highlighted with bold outlines, and destination containers are colored in purple. (Left) All available orders are transferred to the inventory. (Middle) All available orders are injected into the corresponding buffer chunk of its successor fulfillment unit. (Right) Same as the middle operation.

## 4.4 Fulfillment Unit

A fulfillment unit consists of containers and operations. For instance, a fulfillment unit can be a physical warehouse or a virtual e-commerce platform. Figure 5 shows an example fulfillment unit with 3 containers and 3 operations. The top container serves as an inventory, while the middle and bottom containers function as buffers. At each simulation step, only one operation can be selected. The top operation receives orders from both buffers and then stores them in the inventory. The middle operation gathers orders from all 3 containers, consolidates them, and sends them to its connected buffer chunk. Similarly, the bottom operation collects orders from all 3 containers, combines them, and dispatches them to the associated buffer chunk.

## 4.5 Fulfillment Agent

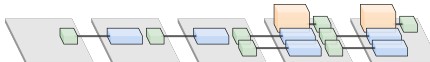

Figure 6: An example fulfillment agent in OFCOURSE, consisting of 5 cascaded fulfillment units.

A fulfillment agent is the decision-making entity in OFCOURSE. In a fulfillment system, the concatenation of all fulfillment units along an order route is defined as a fulfillment agent.

**Action Space:** The action space of a fulfillment agent consists of a list of discrete actions from its constituent fulfillment units. Specifically, fulfillment unit $i$ has a finite set of actions, denoted as $\mathcal{A}_i = \{a_1, \cdots, a_{n_i}\}$, meaning it has a *Discrete* action space of size $|\mathcal{A}_i| = n_i$. For a fulfillment agent composed of $m$ constituent fulfillment units, its action space is the Cartesian product of the action spaces of each constituent fulfillment unit, expressed as $\mathcal{A}_1 \times \ldots \times \mathcal{A}_m$. Consequently, this fulfillment agent has a *MultiDiscrete* [Brockman et al., 2016] action space of size $|\mathcal{A}_1 \times \ldots \times \mathcal{A}_m| = \prod_{i=1}^{m} n_i$. As an example, consider the fulfillment agent represented in Figure 6, whose action space has a shape of [1, 1, 2, 3, 1]. In this case, the fulfillment unit with 1 operation has a dummy action, indicating that it can only select this specific operation in this fulfillment unit. Meanwhile, the fulfillment unit with 2 operations and the fulfillment unit with 3 operations provide multiple candidates to choose from.

**Observation Space:** The observation space of a fulfillment agent combines the observation spaces of its constituent fulfillment units. This *Box* [Brockman et al., 2016] observation space can be represented as $o^{\text{fa}} = [o_1^{\text{fu}} + \cdots + o_m^{\text{fu}}]$, where $o^{\text{fa}}$ refers to the observation of the fulfillment agent, $o_i^{\text{fu}}$ represents the observation of the $i^{th}$ fulfillment unit, and $+$ is the concatenation operator. For the $i^{th}$ fulfillment unit, its observation consists of the combined information from all the containers and operators it includes, which is expressed as $o_i^{\text{fu}} = [o_i^{\text{ctr}} + o_i^{\text{op}}]$, where $o_i^{\text{ctr}}$ indicates the observation of the containers and $o_i^{\text{op}}$ denotes the observation of the operations for the $i^{th}$ fulfillment unit. To ensure flexibility, the observation of an operation $o_i^{\text{op}}$ includes only its price and time. On the other hand, the observation of a container $o_i^{\text{ctr}}$ includes the status of all the orders it carries, capturing the cumulative prices and time.

**Reward Function:** The reward reflects the goal of OF, which is to minimize the total cost of fulfilling all orders within the promised time. At simulation step $t$, we calculated the instant financial cost of the $k^{th}$ fulfillment agent as $c_t^k = \sum_{i=1}^{m_k} \sum_{j=1}^{n_{ik}} c_t^{ijk}$, where $m_k$ represents the number of fulfillment units, $n_{ik}$ signifies the number of orders in the $i^{th}$ fulfillment unit, and $c_t^{ijk}$ denotes the cost of the $j^{th}$ order in the $i^{th}$ fulfillment unit of the $k^{th}$ fulfillment agent. Accordingly, we define the step reward for the $k^{th}$ fulfillment agent as

$$r_t^k = \begin{cases} -c_t^k, & \text{for } t \neq h, \\ -c_t^k - \sum_{i=1}^{m_k} \sum_{j=1}^{n_{ik}} p^{ijk} d^{ijk}, & \text{for } t = h, \end{cases}$$

where $t = h$ represents the last simulation step, $p^{ijk}$ is the penalty coefficient for an unfulfilled order, and $d^{ijk}$ equals to 1 for a failed order and 0 for a successful order.

**Transition Function:** Following each simulation step, the states of all orders are updated based on the transition function. Orders that are currently being operated are transferred to their designated positions in the target containers. In buffers, orders are moved from the current chunk, $B_{T+i}$, to

the next one, $B_{T+i-1}$. In inventories, orders are charged recurring fees in accordance with the quote. Meanwhile, failed orders are handled separately, either by being refunded or fulfilled with priority, with all associated costs being covered by the fulfillment company. Since failed orders cause significantly higher costs, fulfillment agents are incentivized to learn how to avoid them. The implementation of the state transition logic is elucidated in detail with code snippets in Appendix D.

### 4.6 Fulfillment System Overview

As illustrated in Figure 2, a fulfillment system can be conceptually represented as a directed graph, where vertices denote *containers* and directed edges represent *operations*. Containers and operations that are geographically or procedurally adjacent are grouped together into a single *fulfillment unit*. Based on this grouping, all fulfillment units along a fulfillment route are concatenated to form an individual *fulfillment agent*. The objective for the collaborative team of fulfillment agents is to transfer orders from source to destination containers at minimum cost while meeting promised timelines. At each simulation step, a fulfillment agent's action includes the operations of all its constituent fulfillment units. The joint action of all fulfillment agents thereby determines the allocation of *resource* in shared containers.

**As a Simulator for Practical Order Fulfillment:** In practical scenarios, the fulfillment of an individual order often requires multiple transfers between various locations, both geographically and procedurally. In the OFCOURSE framework, these locations are represented as containers, and the transfers themselves are represented as operations. To capture the synergies that exist across multiple locations, containers that are highly correlated are aggregated into fulfillment agents. To model the diverse range of real-world locations and transfer actions, we focus on recording the metrics of prices and time, allowing for a comprehensive representation of the operational details associated with different types of fulfillment units.

**As a Benchmark for Studying MARL Algorithms:** The OFCOURSE framework can serve as a benchmark for studying MARL in heterogeneous systems, while also providing insights potentially useful for real-world applications. In OFCOURSE, fulfillment agents with different observation and action spaces collaborate to minimize the overall cost. This heterogeneity allows for the evaluation of MARL algorithms in non-identical as well as decentralized settings. Alternatively, when agent homogeneity is imposed, OFCOURSE enables the modeling of order fulfillment as a mean-field game [Lasry and Lions, 2007, Guo et al., 2019, Subramanian and Mahajan, 2019]. The objective of cost minimization holds practical relevance since the findings from simulations can provide valuable information for real-world applications.

## 5 Experiment

In this section, based on the computational experiments, we show the capability of OFCOURSE in customizing OF tasks in different contexts, as well as the effectiveness of collectively solving a series of decision problems using MARL based on OFCOURSE.

### 5.1 Various Order Fulfillment Tasks

As our modules cover all the stages and resources that an OF system may involve, various customized fulfillment systems can be constructed using our proposed OFCOURSE framework. We show two examples as follows.

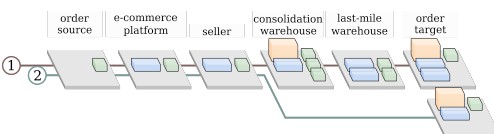

Figure 7: Task 1 — fulfillment of physical and virtual orders in one system by OFCOURSE.

**Task 1 — Fulfillment of Physical and Virtual Orders in One System.** Orders can be physical or virtual. Taking music consumption for example, consumers can either order a vinyl record or download its digital version. As Figure 7 illustrates, agent 1 fulfills physical orders while agent 2

fulfills virtual orders. Both agents share the fulfillment stages of online order processing and seller notification. Excepting the shared stages, agent 1 has extra fulfillment stages for order transportation, indicating its orders have to go through the consolidation warehouse and last-mile warehouse. In the consolidation warehouse, the ordered items can be stored and consolidated with later ones. By performing order consolidation, multiple ordered items can be sent in a single shipment, which cuts overall cost because shipment fees are normally much higher than storage fees. With respect to the two extra stages of agent 1, it has to choose a 3PL company from a finite list of candidates, where each candidate is abstracted as an operation. Usually, no decision has to be made in fulfilling virtual orders. Consequently, each fulfillment unit in agent 2 only contains a single operation.

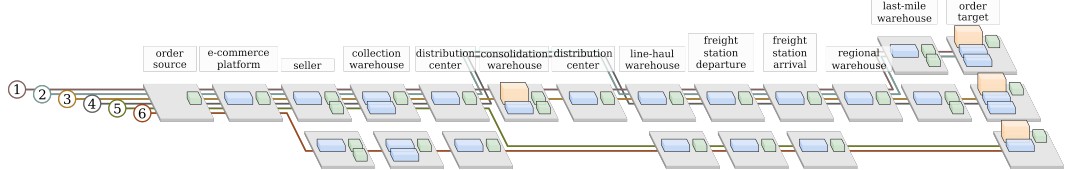

Figure 8: Task 2 — cross-border order fulfillment by OFCOURSE.

**Task 2 — Cross-Border Order Fulfillment.** Cross-border order fulfillment is characterized by its extended and diverse fulfillment stages. Figure 8 shows a cross-border fulfillment system consisting of 6 agents. A customer places an order on the online e-commerce platform, right after which the seller is notified for item collection. This ordered item is then gathered at the nearest collection warehouse and its next station is determined at its associated distribution center. The ordered item can either be sent directly to a line-haul warehouse or go through an extra stage of being consolidated with other items. Next, this ordered item is sent from the departure country to the destination country. Finally, the ordered item is delivered to the door of the customer, which might go through extra stages of storage for cost reduction.

## 5.2 Effectiveness of MARL with OFCOURSE

To show the effectiveness of MARL with OFCOURSE, we conduct experiments to compare an MARL algorithm with three baseline algorithms on the OF tasks built upon OFCOURSE in Section 5.1. In Section 5.2.1, we briefly describe the MARL method and three baseline methods. The experimental results are discussed in Section 5.2.2. Algorithm details can be found in Appendix A and the experimental setup is provided in Appendix B.

### 5.2.1 Baseline

Based on the OF environments defined in Section 5.1, we implement three baseline methods for comparison along with an MARL algorithm:

**Combination of local optima (CLO)** [Gendreau et al., 2010] is a non-learning method. It finds the optimal local action at each fulfillment unit based on the current local observation. Then it combines all these local actions into a global joint action.

**Proximal policy optimization (PPO)** [Schulman et al., 2017, Yu et al., 2022] is an RL method. A gentle introduction to PPO is provided in Appendix A.2. PPO treats the multi-agent system as a single agent, only considering the joint observation-action space. Then it applies the single-agent RL algorithm to this integrated agent.

**Independent PPO (IPPO)** [Tan, 1993, Foerster et al., 2018] is an RL method as well. The algorithmic implementation of IPPO is identical to PPO, while IPPO factors the joint observation-action space into local spaces then treats them as completely independent agents. Then it uses standalone neural networks for each individual agent.

**Heterogeneous agent PPO (HAPPO)** [Kuba et al., 2022] is a theoretically-justified MARL method that supports heterogeneous agents with non-shared policies. HAPPO follows the paradigm of Centralized Training and Decentralized Execution (CTDE) [Kraemer and Banerjee, 2016], allowing for collective learning and coordination during training while still maintaining flexibility and adaptability during execution. The details of HAPPO are elaborated in Appendix A.3.

### 5.2.2 Result

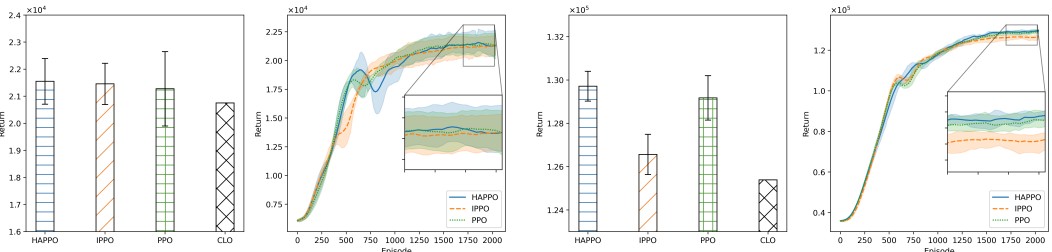

(a) Task 1 — fulfillment of physical and virtual orders in one system.

(b) Task 2 — cross-border order fulfillment.

Figure 9: Experimental results. In each subplot, the bar chart compares the converged performance, and the line chart illustrates the training curves of the learning-based methods.

**Task 1 Results.** The experimental results on Task 1 are illustrated in Figure 9a. In general, the converged performance of HAPPO stays in the vicinity of IPPO, both of which exceed the performance of PPO and CLO. The similar performance of HAPPO and IPPO can be attributed to the fact that agent 1 shares no resource with agent 2, so one agent's observation is redundant information for another agent. For PPO, agent 1 might mistake the coincidental correlation between its reward and the observation of agent 2 as causation, causing its lower return compared to HAPPO and IPPO. As for CLO, it always selects the local operation with the lowest cost without considering the overall operation time required, resulting in inferior performance.

**Task 2 Results.** Figure 9b shows the experimental results on Task 2. Overall, the results demonstrate that HAPPO outperforms the other three methods. Specifically, PPO achieves performance comparable to HAPPO upon convergence, surpassing both IPPO and CLO by a visible margin. It is worth noting that HAPPO and PPO have access to the global state during training, while IPPO and CLO are limited to local observations. The marginally lower performance of PPO compared to HAPPO can potentially be explained by PPO agents erroneously inferring spurious correlations between their own rewards and observations of unrelated agents. On the other hand, the converged performance of IPPO falls between HAPPO/PPO and CLO, which is likely because IPPO agents overlook useful information contained in observations from agents that share the same resources. Similarly, CLO exhibits the worst performance among all the approaches tested. As discussed for Task 1, this can be attributed to CLO optimizing locally without considering the global cost.

## 6 Conclusion

In this paper, we presented Order Fulfillment COoperative mUlti-agent Reinforcement learning Scalable Environment (OFCOURSE), which was the first simulated environment that facilitated solving the complete order fulfillment problem by multi-agent reinforcement learning algorithms. Our computational experiments using OFCOURSE indicated that the joint policy learned by multi-agent reinforcement learning algorithms outperformed the combination of locally optimal policies. We hope that our promising results could motivate more academics and industrial practitioners to approach the complete order fulfillment problem using multi-agent reinforcement learning techniques.

## Acknowledgements

This work is supported by the National Natural Science Foundation of China (No. 72201245) and the Zhejiang Province Postdoctoral Research Fund for Excellent Candidates Project (No. ZJ2023148).

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
