# A  Algorithm

To address the complete Order Fulfillment (OF) problem by Multi-Agent Reinforcement Learning (MARL), in addition to an environment for the agent team to interact with, we still require an algorithm to direct and manage the interaction. Accordingly, we present the MARL algorithm in this section. This section consists of three parts, with each subsequent part building upon the previous one. Appendix A.1 covers the fundamentals of RL, where the actor-critic method is introduced. Appendix A.2 describes the RL algorithm for a single fulfillment agent, which is the proximal policy optimization (PPO) [Schulman et al., 2017]. Appendix A.3 presents the MARL algorithm for the entire fulfillment agent team, which is developed based on the heterogeneous-agent proximal policy optimization (HAPPO) [Kuba et al., 2022].

## A.1  Reinforcement Learning Basics — Actor-Critic

RL covers a multitude of sequential decision-making methods that reward desired behaviors and punish undesired behaviors [Szepesvári, 2010, Sutton and Barto, 2018]. In general, RL methods can be categorized into two groups: value-based methods and policy-based methods. Currently, policy-based methods [Deisenroth et al., 2013] are prevalent because they are compatible with stochastic policies and continuous action spaces. The majority of existing policy-based methods are developed based on the vanilla policy gradient method [Williams, 1992]. Specifically, a policy gradient method updates the policy parameter $\theta$ along the ascending direction of the objective function $J(\pi_\theta)$, namely the policy gradient $\nabla_\theta J(\pi_\theta)$. For the vanilla policy gradient method, the policy gradient $\nabla_\theta J(\pi_\theta)$ is estimated by averaging samples from multiple collected trajectories as

$$\nabla_\theta J(\pi_\theta) = \frac{1}{Bh} \sum_{i=1}^{B} \sum_{t=0}^{h} R(s,a) \nabla_\theta \log \pi_\theta(a|s), \tag{1}$$

where the policy $\pi_\theta(a|s)$ (abbreviated as $\pi(a|s)$ in the following parts) is a parameterized function that outputs a probability distribution over action $a$ given state $s$, $B$ denotes the batch size of trajectories, $h$ is the trajectory length, and $R(s,a)$ represents the reward function that gives a scalar reward based on state $s$ and action $a$. Nevertheless, the variance in vanilla policy gradient is notoriously high. Accordingly, several methods were developed to reduce the variance in policy gradient, among which using a baseline function was proven to be effective. Among these methods, the actor-critic method uses a state-action value function $Q_\pi(s,a)$ to approximate the reward function $R(s,a)$ and a state value function $V_\pi(s)$ as the baseline function $b(s)$, which reduces the variance by introducing tolerable bias. The difference between the state-action value function $Q_\pi(s,a)$ and the state value function $V_\pi(s)$ is called the advantage of action $a$ on state $s$, given by $A_\pi(s,a) = Q_\pi(s,a) - V_\pi(s)$. Incorporating the notation of advantage, the policy gradient of the actor-critic method becomes

$$
\begin{aligned}
\nabla J(\pi) =& \frac{1}{Bh} \sum_{i=1}^{B} \sum_{t=0}^{h} \Big( R(s,a) - b(s) \Big) \nabla \log \pi(a|s) \\
=& \frac{1}{Bh} \sum_{i=1}^{B} \sum_{t=0}^{h} \Big( Q_\pi(s,a) - V_\pi(s) \Big) \nabla \log \pi(a|s) \\
=& \frac{1}{Bh} \sum_{i=1}^{B} \sum_{t=0}^{h} A_\pi(s,a) \nabla \log \pi(a|s).
\end{aligned}
\tag{2}
$$

With each batch of samples, the policy parameter can be updated by

$$\theta_{k+1} \leftarrow \theta_k + \alpha \nabla_{\theta_k} J(\pi_{\theta_k}), \tag{3}$$

where $\theta_k$ and $\theta_{k+1}$ are policy parameters before and after the update, and $\alpha$ denotes the learning rate.

## A.2  Single Agent Reinforcement Learning — PPO

Since the MARL algorithm used in this paper is a multi-agent extension of the single-agent actor-critic method called proximal policy optimization (PPO) [Schulman et al., 2017], we describe PPO here

and expound its multi-agent version in the next part. To be concise, PPO is a first-order trust region method [Kakade and Langford, 2002, Schulman et al., 2015] that improves training stability by constraining the step size within a trust region, avoiding excessively large step sizes which result in detrimental behaviors that are irrecoverable. To formalize PPO, we use $\pi$ to denote the current policy and $\bar{\pi}$ to represent the new policy. Then we define a surrogate objective function which forms a pessimistic bound on the performance of the policy as

$$L_\pi(\bar{\pi}) = J(\pi) + \mathbb{E}_{s \sim \rho_\pi, a \sim \bar{\pi}}[A_\pi(s, a)]. \tag{4}$$

Besides, we express the distance between the distributions of the current policy $\pi$ and the new policy $\bar{\pi}$ using the maximum Kullback-Leibler (KL) divergence [MacKay, 2003] which is

$$D_{KL}^{max}(\pi, \bar{\pi}) = \max_s D_{KL}\Big(\pi(\cdot|s), \bar{\pi}(\cdot|s)\Big). \tag{5}$$

On this basis, we have the following bound that holds,

$$J(\bar{\pi}) \geq L_\pi(\bar{\pi}) - C \cdot D_{KL}^{max}(\pi, \bar{\pi}), \tag{6}$$

where $C = \frac{4\gamma \max_{s,a} |A_\pi(s,a)|}{(1-\gamma)^2}$. As the distance between the current policy $\pi$ and the new policy $\bar{\pi}$ decreases, the surrogate objective $L_\pi(\bar{\pi})$ approaches the original objective $J(\bar{\pi})$. Accordingly, the update at iteration $k + 1$ can be expressed as

$$\pi_{k+1} \leftarrow \arg\max_\pi \Big( L_{\pi_k}(\pi) - C \cdot D_{KL}^{max}(\pi_k, \pi) \Big), \tag{7}$$

where monotonic improvement $J(\pi_{k+1}) \geq J(\pi_k)$ is guaranteed. In practice, using the recommended penalty coefficient $C$ results in excessively small step sizes which further lead to unbearably slow convergence. To overcome this limitation, we take larger update steps using a constraint $\delta$ on the KL divergence between the current policy $\pi$ and the new policy $\bar{\pi}$, namely a trust region constraint, by

$$\theta_{k+1} \leftarrow \arg\max_\theta L_{\pi_{\theta_k}}(\pi_\theta) \text{ s.t. } \mathbb{E}_{s \sim \rho_{\pi_{\theta_k}}}[D_{KL}(\pi_{\theta_k}, \pi_\theta)] \leq \delta. \tag{8}$$

In other words, we optimize $\pi_{\theta_{k+1}}$ to maximize $L_{\pi_{\theta_k}}(\pi_\theta)$ within the boundary specified by $\delta$. To further reduce the cost in computing $\mathbb{E}_{s \sim \rho_{\pi_{\theta_k}}}[D_{KL}(\pi_{\theta_k}, \pi_\theta)]$ in Equation (8), we approximate it by only using the first-order derivatives and optimize the policy parameter $\theta_{k+1}$ by maximizing the clipped surrogate objective [Schulman et al., 2017],

$$L_{\pi_{\theta_k}}(\pi_\theta) = \mathbb{E}_{s \sim \rho_{\pi_{\theta_k}}, a \sim \pi_{\theta_k}} \Big[ \min \Big( \frac{\pi_\theta(a|s)}{\pi_{\theta_k}(a|s)} A_{\pi_{\theta_k}}(s, a), \Xi \big( \frac{\pi_\theta(a|s)}{\pi_{\theta_k}(a|s)} A_{\pi_{\theta_k}}(s, a), 1 \pm \epsilon \big) \Big) \Big], \tag{9}$$

where $\epsilon$ denotes the clipping coefficient and $\Xi(\cdot)$ represents the clipping function that replaces the ratio $\frac{\pi_\theta(a|s)}{\pi_{\theta_k}(a|s)}$ with $1 + \epsilon$ or $1 - \epsilon$ when this ratio exceeds the clipping threshold. Consequently, we update the policy parameter using the policy gradient of this clipped surrogate objective from Equation (9) as

$$\theta_{k+1} \leftarrow \theta_k + \alpha \nabla_{\theta_k} L_{\pi_{\theta_k}}(\pi_{\theta_k}). \tag{10}$$

### A.3 Multi-Agent Reinforcement Learning — HAPPO

Based on the previous parts, we extend the clipped surrogate objective of Equation (9) from the single-agent setting to the multi-agent setting [Kuba et al., 2022]. Following the notations in Section 3.2, we define an ordered subset $i_{1:m} = \{i_1, \ldots, i_m\} \in \mathcal{I}$ and its complement $-i_{i:m}$. In $i_{1:m}$, we denote the $k^{th}$ agent by $i_k$. Then the multi-agent state-action value function can be written as

$$Q_{\boldsymbol{\pi}}^{i_{1:m}}(s, \boldsymbol{a}^{i_{1:m}}) = \mathbb{E}_{\boldsymbol{a}^{-i_{1:m}} \sim \boldsymbol{\pi}^{-i_{1:m}}} \Big[ Q_{\boldsymbol{\pi}}(s, \boldsymbol{a}^{i_{1:m}}, \boldsymbol{a}^{-i_{1:m}}) \Big]. \tag{11}$$

Correspondingly, the multi-agent advantage function for the disjoint sets $j_{1:k}$ and $i_{1:m}$ are

$$A_{\boldsymbol{\pi}}^{i_{1:m}}(s, \boldsymbol{a}^{j_{1:k}}, \boldsymbol{a}^{i_{1:m}}) = Q_{\boldsymbol{\pi}}^{j_{1:k}, i_{1:m}}(s, \boldsymbol{a}^{j_{1:k}}, \boldsymbol{a}^{i_{1:m}}) - Q_{\boldsymbol{\pi}}^{j_{1:k}}(s, \boldsymbol{a}^{j_{1:k}}). \tag{12}$$

Besides, we use $\boldsymbol{\pi} = (\pi^1, \ldots, \pi^n)$ to represent the current joint policy, and use $\bar{\boldsymbol{\pi}} = (\bar{\pi}^1, \ldots, \bar{\pi}^n)$ to denote the new joint policy. Hence, the joint advantage function can be represented as the sum of the local advantages of agents by

$$A_{\boldsymbol{\pi}}^{i_{1:m}}(s, \boldsymbol{a}^{i_{1:m}}) = \sum_{j=1}^{m} A_{\boldsymbol{\pi}}^{i_j}(s, \boldsymbol{a}^{i_{1:j-1}}, a^{i_j}). \tag{13}$$

In Equation (13), agents take actions sequentially following an arbitrary order $i_{1:n}$. For agent $i_1$, it takes an action $\bar{a}^{i_1}$ such that $A^{i_1}(s, \bar{a}^{i_1}) > 0$. For a remaining agent $i_m$ where $m \in [2, n]$, it takes an action $\bar{a}^{i_m}$ such that $A^{i_m}(s, \bar{\boldsymbol{a}}^{i_{1:m-1}}, \bar{a}^{i_m}) > 0$. Consequently, the performance of the agent team is guaranteed to improve since $A_{\boldsymbol{\pi_\theta}}(s, \bar{\boldsymbol{a}})$ is positive. Analogous to the definition of the joint policy, we define $\bar{\boldsymbol{\pi}}^{i_{1:m-1}} = \prod_{j=1}^{m-1} \bar{\pi}^{i_j}$ as some other joint policy of agents $i_{1:m-1}$. Additionally, we define $\hat{\pi}^{i_m}$ as some other policy of agent $i_m$. On this basis, we have the surrogate objective,

$$L_{\boldsymbol{\pi}}^{i_{1:m}}(\bar{\boldsymbol{\pi}}^{i_{1:m-1}}, \hat{\pi}^{i_m}) = \mathbb{E}_{s \sim \rho_{\boldsymbol{\pi}}, \boldsymbol{a}^{i_{1:m-1}} \sim \bar{\boldsymbol{\pi}}^{i_{1:m-1}}, a^{i_m} \sim \hat{\pi}^{i_m}}[A_{\boldsymbol{\pi}}^{i_m}(s, \boldsymbol{a}^{i_{1:m-1}}, \boldsymbol{a}^{i_m})]. \tag{14}$$

Meanwhile, for any joint policy $\bar{\boldsymbol{\pi}}$, we have the following inequality holds that

$$J(\bar{\boldsymbol{\pi}}) \geq J(\boldsymbol{\pi}) + \sum_{m=1}^{n} [L_{\boldsymbol{\pi}}^{i_{1:m}}(\bar{\boldsymbol{\pi}}^{i_{1:m-1}}, \bar{\pi}^{i_m}) - C \cdot D_{KL}^{max}(\pi^{i_m}, \bar{\pi}^{i_m})]. \tag{15}$$

Concisely, we sequentially update the policy of each agent, and the optimization objective of each agent incorporates the updates of all previous ones. At iteration $k + 1$, given a permutation of agents $i_{1:n}$, agent $i_m \in \{1, \ldots, n\}$ sequentially optimize the policy parameter $\theta_{k+1}^{i_m}$ by maximizing the constrained objective,

$$\theta_{k+1}^{i_m} \leftarrow \arg\max_{\theta^{i_m}} \mathbb{E}_{s \sim \rho_{\boldsymbol{\pi_{\theta_k}}}, \boldsymbol{a}^{i_{1:m-1}} \sim \boldsymbol{\pi}_{\theta_{k+1}^{i_{1:m-1}}}^{i_{1:m-1}}, a^{i_m} \sim \pi_{\theta^{i_m}}^{i_m}} [A_{\boldsymbol{\pi_{\theta_k}}}^{i_m}(s, \boldsymbol{a}^{i_{1:m-1}}, a^{i_m})]$$

$$\text{s.t. } \mathbb{E}_{s \sim \rho_{\boldsymbol{\pi_{\theta_k}}}}[D_{KL}(\pi_{\theta_k^{i_m}}^{i_m}(\cdot|s), \pi_{\theta^{i_m}}^{i_m}(\cdot|s))] \leq \delta. \tag{16}$$

Then we apply a linear approximation to the objective function and a quadratic approximation to the KL constraint. Accordingly, the closed-form update for the optimization problem can be expressed as

$$\theta_{k+1}^{i_m} \leftarrow \theta_k^{i_m} + \eta^j \sqrt{\frac{2\delta}{\boldsymbol{g}_k^{i_m}(\boldsymbol{H}_k^{i_m})^{-1}\boldsymbol{g}_k^{i_m}}} (\boldsymbol{H}_k^{i_m})^{-1}\boldsymbol{g}_k^{i_m}, \tag{17}$$

where $\boldsymbol{H}_k^{i_m} = \nabla_{\theta^{i_m}}^2 \mathbb{E}_{s \sim \rho_{\boldsymbol{\pi_{\theta_k}}}}[D_{KL}(\pi_{\theta_k^{i_m}}^{i_m}(\cdot|s), \pi_{\theta^{i_m}}^{i_m}(\cdot|s))]|_{\theta^{i_m} = \theta_k^{i_m}}$ is the Hessian of the expected KL divergence, $\boldsymbol{g}_k^{i_m}$ is the gradient of the objective, $\eta^j < 1$ is a positive coefficient that is found by backtracking line search [Bertsekas, 1997], and the product of $(\boldsymbol{H}_k^{i_m})^{-1}\boldsymbol{g}_k^{i_m}$ can be computed by the conjugate gradient algorithm [Shewchuk, 1994]. For each state $s$, we have

$$\mathbb{E}_{\boldsymbol{a}^{i_{1:m-1}} \sim \bar{\boldsymbol{\pi}}^{i_{1:m-1}}, a^{i_m} \sim \hat{\pi}^{i_m}}[A_{\boldsymbol{\pi}}^{i_m}(s, \boldsymbol{a}^{i_{1:m-1}}, a^{i_m})] = \mathbb{E}_{\boldsymbol{a} \sim \boldsymbol{\pi}}\left[\left(\frac{\hat{\pi}^{i_m}(a^{i_m}|s)}{\pi^{i_m}(a^{i_m}|s)} - 1\right)\frac{\bar{\boldsymbol{\pi}}^{i_{1:m-1}}(\boldsymbol{a}^{i_{1:m-1}}|s)}{\boldsymbol{\pi}^{i_{1:m-1}}(\boldsymbol{a}^{i_{1:m-1}}|s)}A_{\boldsymbol{\pi}}(s, \boldsymbol{a})\right] \tag{18}$$

We estimate the advantage function

$$\mathbb{E}_{\boldsymbol{a}^{i_{1:m-1}} \sim \boldsymbol{\pi}_{\theta_{k+1}^{i_{1:m-1}}}^{i_{1:m-1}}, a^{i_m} \sim \pi_{\theta^{i_m}}^{i_m}}\left[A_{\boldsymbol{\pi_{\theta_k}}}^{i_m}(s, \boldsymbol{a}^{i_{1:m-1}}, a^{i_m})\right] \tag{19}$$

with the estimator

$$\left(\frac{\pi_{\boldsymbol{\theta}}^{i_m}(a^{i_m}|s)}{\pi_{\boldsymbol{\theta}_k}^{i_m}(a^{i_m}|s)} - 1\right)M^{i_{1:m}}(s, \boldsymbol{a}), \tag{20}$$

where $M^{i_{1:m}} = \frac{\bar{\boldsymbol{\pi}}^{i_{1:m-1}}(\boldsymbol{a}^{i_{1:m-1}}|s)}{\boldsymbol{\pi}^{i_{1:m-1}}(\boldsymbol{a}^{i_{1:m-1}}|s)}\hat{A}(s, \boldsymbol{a})$. Specifically, we employ the generalized advantage estimation (GAE) [Schulman et al., 2016] as the value estimator, which actively trades off variance and bias,

$$\hat{A}_t^{\kappa,\lambda} = \sum_{l=0}^{h}(\kappa\lambda)^l\left(-V_\pi(s_{t+l}) + r_{t+l} + \kappa V_\pi(s_{t+l+1})\right), \tag{21}$$

where $\kappa$ and $\lambda$ are coefficients that control the variance-bias trade-off. By only considering the first order derivatives, the computational load of $\boldsymbol{H}_k^{i_m}$ is significantly reduced. At step $k + 1$, agent $i_m$ selects the policy parameter $\theta_{k+1}^{i_m}$ to maximize the clipped surrogate objective

$$L_{\boldsymbol{\pi_{\theta_k}}}(\pi_{\theta^{i_m}}^{i_m}) = \mathbb{E}_{s \sim \rho_{\boldsymbol{\pi_{\theta_k}}}, \boldsymbol{a} \sim \boldsymbol{\pi_{\theta_k}}}\left[\min\left(\frac{\pi_{\theta^{i_m}}^{i_m}(a^i|s)}{\pi_{\theta_k^{i_m}}^{i_m}(a^i|s)}M^{i_{1:m}}(s, \boldsymbol{a}), \Xi\left(\frac{\pi_{\theta^{i_m}}^{i_m}(a^i|s)}{\pi_{\theta_k^{i_m}}^{i_m}(a^i|s)}, 1 \pm \epsilon\right)M^{i_{1:m}}(s, \boldsymbol{a})\right)\right], \tag{22}$$

where $\epsilon$ represents the clipping coefficient and $\Xi(\cdot)$ denotes the clipping function that replaces the ratio $\frac{\pi^{i_m}_{\theta^{i_m}}(a^i|s)}{\pi^{i_m}_{\theta^{i_m}_k}(a^i|s)}$ with $1 + \epsilon$ or $1 - \epsilon$ when this ratio exceeds the clipping threshold. Consequently, we update the policy parameter of agent $i_m$ by

$$\theta^{i_m}_{k+1} \leftarrow \theta^{i_m}_k + \alpha \nabla_\theta L_{\pi_{\theta_k}}(\pi^{i_m}_{\theta^{i_m}}), \tag{23}$$

where $\theta^{i_m}_k$ and $\theta^{i_m}_{k+1}$ denote the policy parameters of agent $i_m$ before and after the update, and $\alpha$ is the learning rate. To sum up, the complete procedure is given in Algorithm 1.

---

**Algorithm 1** Heterogeneous Multi-Agent Reinforcement Learning for Order Fulfillment.

1: Initialize the joint policy $\pi_{\theta_0} = (\pi^1_{\theta^1_0}, \ldots, \pi^n_{\theta^n_0})$ and the GAE value network $V_w$.
2: **for** $k = 0, 1, \ldots, K - 1$ **do**
3:      Sample trajectories using the joint policy $\pi_{\theta_k} = (\pi^1_{\theta^1_k}, \ldots, \pi^n_{\theta^n_k})$.
4:      Store transition samples $\{(o^i_t, a^i_t, o^i_{t+1}, r_t), \forall i \in \mathcal{I}, t \in [0, h]\}$ in the replay buffer.
5:      Calculate the advantage $\hat{A}(s, \boldsymbol{a})$ using the critic network.
6:      Generate a random sequence for agents $i_{i:n}$.
7:      Let $M^{i_1}(s, \boldsymbol{a}) \leftarrow \hat{A}(s, \boldsymbol{a})$.
8:      **for** agent $i_m = i_1, \ldots, i_n$ **do**
9:          $\theta^{i_m}_{k+1} \leftarrow \frac{1}{Bh} \sum_{j=1}^{B} \sum_{t=0}^{h} \min \left( \frac{\pi^{i_m}_{\theta^{i_m}}(a^{i_m}_t|o^{i_m}_t)}{\pi^{i_m}_{\theta^{i_m}_k}(a^{i_m}_t|o^{i_m}_t)} M^{i_{i:m}}(s_t, \boldsymbol{a}_t), clip\left( \frac{\pi^{i_m}_{\theta^{i_m}}(a^{i_m}_t|o^{i_m}_t)}{\pi^{i_m}_{\theta^{i_m}_k}(a^{i_m}_t|o^{i_m}_t)}, 1 \pm \epsilon \right) M^{i_{i:m}}(s_t, \boldsymbol{a}_t) \right)$.
10:          $M^{i_{i:m+1}}(s, \boldsymbol{a}) \leftarrow \frac{\pi^{i_m}_{\theta^{i_m}_{k+1}}(a^{i_m}_t|o^{i_m}_t)}{\pi^{i_m}_{\theta^{i_m}_k}(a^{i_m}_t|o^{i_m}_t)} M^{i_{i:m}}(s, \boldsymbol{a}) \quad \forall m = n$.
11:      **end for**
12:      Update $w_{k+1}$ for the critic network by $\arg\min_w \frac{1}{Bh} \sum_{j=1}^{B} \sum_{t=0}^{h} \left( V_w(s_t) - \hat{R}_t \right)^2$.
13: **end for**

---

## B  Experimental Setup

Regarding neural network architectures, the shape of a critic neural network is $[\texttt{obs}, \texttt{hid}, \texttt{hid}, 1]$ and the shape of an actor neural network is $[\texttt{obs}, \texttt{hid}, \texttt{hid}, \texttt{act}]$. In other words, the actor network and the critic network only differ in the output layer and share the same structure for the remaining 2 hidden layers which are linear layers of size 256 followed by ReLU activation functions [Agarap, 2018]. In terms of the training procedure, the total number of episodes is $128 \times 16 = 2048$, the episode length is 200 steps, and the batch size is 1024. We use the Adam optimizer [Kingma and Ba, 2015] for all neural networks. With respect to the optimizer, we use $3 \times 10^{-3}$ as the learning rate for all actor networks and critic networks. With regard to the advantage estimator, we set the GAE parameters [Schulman et al., 2016] $\kappa = 0.99$ and $\lambda = 0.95$. Besides, the gradient clipping threshold [Schulman et al., 2017] is $\epsilon = 1.0$ for all actor networks and critic networks. In Equation (4.5), the undelivered penalty $p$ is set to 50. All models were trained with 5 random seeds on Intel Xeon Platinum 8396B CPU@2.90GHz and NVIDIA A100-80GB GPU.

## C  Comparison to Existing Benchmarks

To highlight how our proposed benchmark differs from existing approaches focused on sub-tasks of order fulfillment, we compare the objectives, observations, and actions in Table 1. It should be noted that multiple formulations exist for each sub-task. For illustrative purposes, we include representative works on scheduling [Zhang et al., 2020], bin-packing [Duan et al., 2019], inventory management [Oroojlooyjadid et al., 2022], and vehicle routing [James et al., 2019]. This comparison demonstrates key distinctions between modeling isolated sub-problems versus our approach of an integrated fulfillment system benchmark.

Table 1: Comparison to existing benchmarks.

| Benchmark | Objective | Observation | Action |
|---|---|---|---|
| scheduling | makespan | disjunctive graph | operation |
| bin-packing | surface area | bin size sequence | sequencing & orientation |
| inventory management | inventory level | historical quadruple | order quantity |
| vehicle routing | traveling length | node coordinations | node |
| **order fulfillment (ours)** | **financial cost** | **order information** | **operation** |

# D   State Transition

To explicate the state transition logic in OFCOURSE, the following code snippets provide an illustrative implementation. At each simulation step, the joint action is input at the team level. The team then calls its agents and provides each agent's individual action as input. Next, each agent then passes the pertinent unit action as input to its constituent fulfillment units. Within a fulfillment unit, the execute() function takes the unit action and alters the state information for all orders carried by that unit based on preset probabilities corresponding to the operation encoded in the unit action.

In OFCOURSE, each simulation step invokes `Team.step(agents_action)` once, where each agent retrieves its designated `action` from `agents_action` (line 8 in *team.py*).

```
1  class Team:
2      ...
3      def step(self, agents_action):
4          self.step_count += 1
5          _costs = [0 for agent_i in range(self.n_agents)]
6          for agent_i, action in enumerate(agents_action):
7              if not (self.agent_dones[agent_i]):
8                  _costs[agent_i] = self.agents[agent_i].step(action,
       self.step_count)
9          if self.step_count >= self.max_step:
10             for agent_i in range(self.n_agents):
11                 _costs[agent_i] += self.agents[agent_i].clearance(self
       .undelivered_penalty)
12                 self.agent_dones[agent_i] = True
13         _obs = []
14         for agent_i in range(self.n_agents):
15             _obs.append(self.get_agent_obs(agent_i))
16         _rewards = [-_cost + self.args.r_baseline for _cost in _costs]
17         return _obs, _rewards, self.agent_dones, {}
```

Each call to `Agent.step(action, step_count)` first updates order information (line 5 in *agent.py*), then executes the designated `action` (line 6 in *agent.py*). Specifically, `Agent.update()` iterates through all fulfillment units and their carried orders (line 12 in *agent.py*). Similarly, `Agent.execute(action, step_count)` iterates through all units to take the corresponding actions (line 18 in *agent.py*).

```
1  class Agent:
2      ...
3
4      def step(self, action, step_count=None):
5          _reward_update = self.update()
6          _reward_execute = self.execute(action, step_count)
7          return _reward_update + _reward_execute
8
9      def update(self):
10         _step_price = 0
11         for _fulfillment_unit in self.fulfillment_units:
12             _step_price += _fulfillment_unit.update()
13         return _step_price
14
15     def execute(self, action, step_count=None):
```

```
16          _step_price = 0
17          for _agent_i, _fulfillment_unit in enumerate(self.
        fulfillment_units):
18              _step_price += _fulfillment_unit.execute(action[_agent_i],
         step_count)
19          return _step_price
```

From a fulfillment unit's perspective, FulfillmentUnit.update() updates order information (line 10 in *fulfillment_unit.py*), while FulfillmentUnit.execute(action, step_count) carries out order transfers (line 14 in *fulfillment_unit.py*).

```
1  class FulfillmentUnit:
2      def __init__(self, latitude=None, longitude=None):
3          self.containers = deque()
4          self.operations = deque()
5          ...
6
7      def update(self):
8          _step_price = 0
9          for _container in self.containers:
10             _step_price += _container.update()
11         return _step_price
12
13     def execute(self, action, step_count=None):
14         return self.operations[action].execute(step_count)
```