# OpenReview forum: "OFCOURSE: A Multi-Agent Reinforcement Learning Environment for Order Fulfillment"
_NeurIPS.cc/2023/Track/Datasets_and_Benchmarks — NeurIPS 2023 Datasets and Benchmarks Poster_

### Official Review · Reviewer_RuY6 · 2023-07-20
**paper review**

**Rating:** 6
**Confidence:** 4

**Strengths:**

- OFCOURSE is a framework for studying important order fulfillment problems in modern society, and even industry practitioners who do not study reinforcement learning or artificial intelligence can benefit from it.

- While previous works considers the order fulfillment process by dividing it into individual sub-problems, OFCOURSE considers a centralized process. In addition, OFCOURSE also considers the interaction between the agents (orders) by incorporating MARL algorithms, rather than existing machine learning or single-agent reinforcement learning.

- The paper well describes how order fulfillment problem can be seen as a DecPOMDP environment. In addition, the proposed modular design allows efficient customizing of the environment depending on various purposes.

**Additional Feedback:**

There is no additional feedback.

**Clarity:**

The paper is well-written and accessible to readers unfamiliar with order fulfillment.

**Correctness:**

Most claims in the paper seem correct, but there are some questions related to evaluation method/experimental design (Questions are stated above).

**Documentation:**

The source code URL is publicly available to use the OFCOURSE framework, and the experimental setup such as hyperparameters related to reinforcement learning is publicly available too.

**Ethics:**

I think there are no ethical issues.

**Limitations:**

The paper provides appropriate limitations, and it seems that there is no potential negative societal impact.

**Opportunities For Improvement:**

1. The paper effectively demonstrates the significance of OFCOURSE as a solution for the order fulfillment problem. However, it would be beneficial to provide further elaboration to emphasize its importance as a benchmark for studying MARL algorithms. Introducing the advantages of OFCOURSE over existing MARL benchmarks would enhance its significance in the research field.

2. The order fulfillment problem is well explained and easy to understand, but its relation to reinforcement learning seems unclear. It would be helpful to provide a comparison between the reward functions, states, and action spaces in the order fulfillment benchmark introduced in the paper and existing benchmarks. Additionally, explaining how a particular action leads to transitioning from one state to another state with a given probability would enhance the understanding of the problem.

3. There seems to be a scale-up issue and it would be helpful to have some explanation and research on this. For example, when multi agents are incorporated into the OFCOURSE framework, each order is an agent, so if the number of orders increases very large as in reality, the number of agents increases significantly. Consequently, the non-stationary problem also intensifies, leading to an exponential increase in the observation and action space. As a result, the learning process may slow down.

4. The part of the paper where the various reinforcement learning algorithms were experimented with on the OFCOURSE benchmark would be better understood with more details about the experiments. For instance, as explained in the main body of the paper, the reward function is set to be negative, but the return value in the experiment is set to be positive, it would be helpful to include some mention of the scale of the return.

5. Since the problem deals with order fulfillment, it would be beneficial to include experiments that investigate the performance as the number of orders increases. Additionally, including an experiment that measures the comparative improvement and efficiency of the given algorithm in relation to existing research on order fulfillment would provide valuable insights.

**Relation To Prior Work:**

While previous research has considered each order fulfillment problem as a subproblem, this paper is described as the first to consider the complete order fulfillment problem in an integrated manner. The differences are clearly discussed in the paper well.

**Summary And Contributions:**

In order to efficiently proceed with the order fulfillment process, the authors introduce OFCOURSE, a multi-agent reinforcement learning benchmark. Previous studies approach the order fulfillment process using the single-agent reinforcement learning algorithms. Dividing the entire process into sub-problems, the traditional-RL algorithms were used to solve each sub-problem. However, the interdependence of series of online decisions makes the order fulfillment problem difficult to solve. By introducing a multi-agent framework that can perform complete order fulfillment steps in the OpenAI Gym style, the authors test multiple MARL algorithms on OFCOURSE. The experimental results show the superiority of multi-agent approach compared to a combination of locally optimal policies in the order fulfillment problem.

---

> ### Author Response · Authors · 2023-08-19
> **Response to Reviewer RuY6 (Part 1/2)**
>
> First of all, we would like to express our appreciation to Reviewer RuY6 for your detailed comments and suggestions. We respond to your comments and suggestions as follows:
>
> **Comment 1. The paper effectively demonstrates the significance of OFCOURSE as a solution for the order fulfillment problem. However, it would be beneficial to provide further elaboration to emphasize its importance as a benchmark for studying MARL algorithms. Introducing the advantages of OFCOURSE over existing MARL benchmarks would enhance its significance in the research field.**
>
> ***Response.***
>
> Thank you for this valuable comment. The importances of OFCOURSE as a MARL benchmark are threefold:
> - OFCOURSE is the first MARL benchmark to model the complete order fulfillment problem, rather than focusing on individual subproblems.
> - OFCOURSE contributes as a heterogeneous MARL benchmark. By incorporating diverse agent types with distinct action spaces, OFCOURSE enables studying the complex coordination challenges that arise in heterogeneous benchmarks.
> - OFCOURSE is devised on the practical problem of order fulfillment. MARL algorithms developed and insights gained through OFCOURSE can be potentially translated beyond simulation to guide real-world deployments.
>
> According to your suggestions, we have emphasized OFCOURSE's importance as a MARL benchmark in the updated manuscript (lines 35-37, 47-49, 76-79, and 301-308).
>
> **Comment 2. The order fulfillment problem is well explained and easy to understand, but its relation to reinforcement learning seems unclear. It would be helpful to provide a comparison between the reward functions, states, and action spaces in the order fulfillment benchmark introduced in the paper and existing benchmarks.**
>
> ***Response.***
>
> Thank you for this insightful advice. Per your recommendation, we have added a comparison of state space, action space, and objective function between our fulfillment benchmark to existing benchmarks in Table 1 of Appendix C of the updated manuscript. As shown in the table, our benchmark's reward functions, states, and action spaces entirely differs those in the existing benchmarks. We agree with you that this comparison can helps situate our contributions within the literature. Thank you again for this valuable advice.
>
> **Comment 3. Additionally, explaining how a particular action leads to transitioning from one state to another state with a given probability would enhance the understanding of the problem.**
>
> ***Response.***
>
> Thank you for the constructive feedback on elucidating state transitions resulting from actions. Per your recommendation, we have included supplementary code snippets in Appendix D of the updated manuscript to demonstrate how a particular action leads to transitioning from one state to another state with the given probability. For your convenience, we provide a brief overview here:"The joint action is input at the agent-team level. The team then calls its agents and provides each agent's individual action as input. Next, each agent calls its fulfillment units, passing the unit action as input to each unit. Within a fulfillment unit, the execute() function takes the unit action as input and changes the information for all orders carried by this fulfillment unit, based on the preset transition probabilities associated with the corresponding operation."

---

> ### Author Response · Authors · 2023-08-19
> **Response to Reviewer RuY6 (Part 2/2)**
>
> **Comment 4. There seems to be a scale-up issue and it would be helpful to have some explanation and research on this. For example, when multi agents are incorporated into the OFCOURSE framework, each order is an agent, so if the number of orders increases very large as in reality, the number of agents increases significantly. Consequently, the non-stationary problem also intensifies, leading to an exponential increase in the observation and action space. As a result, the learning process may slow down.**
>
> ***Response.***
>
> Thank you for the perceptive observations highlighting potential scale-up challenges with multi-agent reinforcement learning frameworks like OFCOURSE. We appreciate you thoughtfully outlining this significant consideration. As you astutely noted, modeling individual orders as agents could lead to an explosion of the observation and action spaces as volumes increase, consequently intensifying the non-stationarity issue. However, OFCOURSE represents the concatenation of fulfillment units as individual agents, rather than each order separately. Our agent definition aims to facilitate operations over multiple orders, such as order consolidation. With our design, the number of agents scales with the established fulfillment routes, not total order volume. Based on our historical data, the route count hardly exceeds 6, even as volumes grow. In a practical fulfillment system, agents can be divided into mutually exclusive groups (different groups share no resource) so that the team size can be further reduced. However, we concur that studying scalability and non-stationarity in order fulfillment would be undoubtedly meaningful for other scenarios in the future work.
>
> **Comment 5. The part of the paper where the various reinforcement learning algorithms were experimented with on the OFCOURSE benchmark would be better understood with more details about the experiments. For instance, as explained in the main body of the paper, the reward function is set to be negative, but the return value in the experiment is set to be positive, it would be helpful to include some mention of the scale of the return.**
>
> ***Response.***
>
> Thank you for this insightful comment. As you rightly noted, our paper implements the common practice of defining the reward function as the negative cost offset by a configurable baseline reward. The empirical baseline used in our experiments is calculated by multiplying the expected order volume by the per-order penalty to determine the return scale. Additionally, as you mention, the return is often further scaled to a range like [0,1] based on the algorithm needs. Please find the detailed implementation in https://github.com/GitYiheng/ofcourse/blob/main/env/exp1_env.py line 62.
>
> We sincerely hope our detailed responses have sufficiently clarified any misunderstandings and addressed your comments. If you are satisfied with our responses, we would be grateful if you would consider adjusting your review rating accordingly. Please do not hesitate to provide any additional suggestions you may have and we welcome any feedback that could help further improve our work. Our aim is to fully resolve all of your thoughtful comments and questions. Thank you again for your time and valuable insights.

---

> > ### Comment · Reviewer_RuY6 · 2023-08-21
> >
> > Thank you very much for providing detailed responses to the comments first. Additionally, after reviewing the responses and the updated manuscript, my inquiries have been thoroughly addressed. I truly appreciate it. Furthermore, as I have incorporated all the points mentioned in the 'Opportunities For Improvement' into the manuscript, the review rating has been adjusted.

---

> > > ### Author Response · Authors · 2023-08-21
> > > **Response to Response by Reviewer RuY6**
> > >
> > > We appreciate you taking the time to thoroughly review our revised manuscript. Your insightful comments helped us to strengthen this work and we sincerely appreciate you raising the rating.

---

### Official Review · Reviewer_h6Wq · 2023-07-21
**This paper introduces a novel approach to model order fulfillment, offering a comprehensive and flexible environment that covers all stages of the process. However, there are areas within the proposed model that could be further improved.**

**Rating:** 6
**Confidence:** 3
**Clarity:** Well written and easy to follow.

**Strengths:**

1. This paper presents the pioneering simulation environment that comprehensively incorporates all stages of the Order Fulfillment problem.
2. OFCOURSE exhibits flexibility and ease of modification, supporting modular design and step-wise coordination. As a result, it has the potential to be highly adaptable across a wide range of domains.

**Additional Feedback:**

Suggestion for improvement: I noticed that in Task 2, the PPO algorithm outperformed IPPO. While this situation is not unusual, it may be beneficial to conduct a comparative analysis over independent MARL, MARL trained by CTDE, and single-agent RL algorithms. This would further emphasize the advantages of modeling the order fulfillment scenario as a Markov game rather than a Markov process.
Question: How do you perceive the application of this simulator in real-world scenarios, taking into account considerations such as the sim2real gap, observability of agents, and transferability of policies?

**Correctness:**

The claims made in the submission appear to be correct. And it would be better if the authors provide more MARL algorithms to fulfill the benchmark.

**Documentation:**

The provided details for the two sample tasks in the submission are sufficient. However, although the authors claim that the simulator is capable of accommodating various scenarios, they have not provided any relevant tutorials or instructional materials to support this assertion.

**Limitations:**

OFCOURSE's consideration is primarily focused on price and time, while neglecting other details. Consequently, its applicability in practical scenarios or specialized cargo transportation may encounter certain limitations or gaps.

**Opportunities For Improvement:**

1. To enhance the benchmark, the authors may consider incorporating additional order fulfillment scenarios or MARL algorithms. Furthermore, it would be beneficial to include a thorough analysis of the RL and MARL results.
2. As order fulfillment applications serve real-world scheduling purposes eventually, the authors could enrich the discussion by exploring topics such as policy transfer and addressing the gap between simulation and real-world environments, including sim2real challenges. This would contribute to a more comprehensive analysis and discussion of the practical implications of order fulfillment in real-world settings.

**Relation To Prior Work:**

Yes, it is clearly discussed.

**Summary And Contributions:**

This paper introduces OFCOURSE, a novel multi-agent simulation environment that represents the Order Fulfillment (OF) problem as a Markov game. According to the authors, OFCOURSE stands out as the very first environment that comprehensively incorporates full stages of order fulfillment, including order handling, packing and pickup, storage, as well as the last-mile delivery stage. Consequently, this advancement enables the training of decision-making algorithms in the entire process of Order Fulfillment.

---

> ### Author Response · Authors · 2023-08-19
> **Response to Reviewer h6Wq**
>
> We greatly appreciate you dedicating time to thoroughly review our manuscript and share thoughtful feedback. Please find our responses below:
>
> **Comment 1. I noticed that in Task 2, the PPO algorithm outperformed IPPO. While this situation is not unusual, it may be beneficial to conduct a comparative analysis over independent MARL, MARL trained by CTDE, and single-agent RL algorithms. This would further emphasize the advantages of modeling the order fulfillment scenario as a Markov game rather than a Markov process.**
>
> ***Response.***
>
> It is thoughtful of you to point out this issue. We would like to bring to your attention that we have already undertaken the experiments that encompass the comparison of various algorithms, including single-agent RL (PPO), independent MARL (IPPO), and MARL trained by CTDE (HAPPO). This comprehensive comparative analysis is thoroughly presented in Section 5, specifically lines 361-377. As you observe from our experimental results, HAPPO has demonstrated superior performance compared to the other algorithms. This outcome underscores the advantageous nature of adopting the Markov game model over the Markov decision process in order fulfillment scenario. Additionally, we would like to reiterate that our choice to employ the Markov game model was motivated not only by its performance but also by its inherent scalability and adaptability to the dynamic nature of order fulfillment systems, as elaborated in lines 119-129.
>
> **Comment 2. How do you perceive the application of this simulator in real-world scenarios, taking into account considerations such as the sim2real gap, observability of agents, and transferability of policies?**
>
> ***Response.***
>
> We appreciate you raising the important issue of applicability. We acknowledge the ubiquitous challenges of the sim2real gap, partial observability, and transferability of policies faced by all MARL benchmarks and simulators. At the modeling level, we aimed to narrow sim2real gap by focusing on universal metrics of order-wise time and cost that generalize across fulfillment systems. To alleviate the partial observability issue, we implemented HAPPO as a baseline algorithm that follows the centralized training with decentralized execution paradigm.
> While it’s true that the complete closure of the sim2real gap might remain elusive, it’s important to emphasize the value of benchmarks and simulators in guiding practical decision-making and policy research. The applications of our simulator are two-fold:
>
> - Decision-Making Support: Our simulator can provide valuable insights by enabling simulated analysis of existing fulfillment systems. This offers a platform for decision-makers to explore different scenarios and make informed choices.
> - Policy Transfer Research: Our simulator is also conducive to researching policy transfer in order fulfillment scenarios. For instance, it can serve as a foundation for generating initial policies in small-scale real-world deployments through tabula rasa simulations. Subsequently, researchers can delve into techniques like domain adaptation, utilizing our framework as a versatile testing ground.
>
> **Comment 3. OFCOURSE's consideration is primarily focused on price and time, while neglecting other details. Consequently, its applicability in practical scenarios or specialized cargo transportation may encounter certain limitations or gaps.**
>
> ***Response.***
>
> We are appreciative that you have highlighted this issue. The decision to primarily focus on price and time within OFCOURSE was a deliberate choice aimed at capturing the fundamental aspects of common order fulfillment settings. If the "other details" you referred to are constraints, our simulation environment implemented them by action masking, namely temporarily zero out the probabilities of unavailable discrete actions. If the "other details" you referred to are resources, our simulation environment incorporates them as a predefined function. This function takes the number of orders being processed as input and generates a quoted price as output. By incorporating these details and leveraging the practical considerations of price and time, OFCOURSE well-suited to addressing a wide range of real-world scenarios.
>
> We sincerely appreciate your insightful feedback. Thank you again for your time and valued comments. Please advise any areas that would benefit from clarification or expansion. We welcome your expert perspective.

---

### Official Review · Reviewer_HNVp · 2023-07-24
**Government regulations on the platform**

**Rating:** 10
**Confidence:** 4
**Correctness:** I think that the claims made in the s…
**Clarity:** The paper is succinct and well-written.

**Strengths:**

The paper contributes to the literature by setting up a framework on solving the order fufillment problem. This paper devolopes algorithm to solve the problem by machine learning. By modelling OF problem as a mean-field game, the paper also connects an operation research problem to economics. This may stimulate more works in economic research.

**Additional Feedback:**

There is a trend along which researchers connect the mean-field game to reinforcement learning. I think that the authors can mention this point in the paper.

**Documentation:**

The paper is well-organized and the code is posted on Github.

**Ethics:**

I have no concern about ethics with the submission.

**Limitations:**

The authors can also look at this complicated problem from a perspective of dynamic mechanism design. The algorithm of the e-commerce platform is mechanism. The platform is searching an optimal mechanism to minimize the cost.

**Opportunities For Improvement:**

E-commerce is highly regulated in many countries. I suggest the author take into government regulations when simulating the problem. Then the enviorenment in the model is more realistic. Under this situation, the policy functions in the reinforcement learning is more practical.

**Relation To Prior Work:**

The paper clearly states its contributions relative to the literature.

**Summary And Contributions:**

The authors invent a model in which they view the order fulfillment on e-commerce platform as a multi-agent reinforecement learning problem. Different fufillment units in different stages as different agents in the game. The overall aim of the game is to minimize the cost of fulfillments. The interesting part of the paper is to transform a complicated OF problem into a reinforcement learning procedure.

---

> ### Author Response · Authors · 2023-08-19
> **Response to Reviewer HNVp**
>
> We sincerely appreciate you taking the time to thoroughly review our work and provide insightful suggestions for improvement. Your comments have helped strengthen the manuscript's practical applicability and connections to broader research areas. Please find our responses below:
>
> **Comment 1. E-commerce is highly regulated in many countries. I suggest the author take into government regulations when simulating the problem. Then the environment in the model is more realistic. Under this situation, the policy functions in the reinforcement learning is more practical.**
>
> ***Response.***
>
> Thank you for the insightful suggestion to incorporate government regulations into the simulation environment. As you rightly note, including relevant regulatory constraints would enhance the model's realism and practical applicability. Our current approach implements such constraints through action masking, which temporarily zeros out the probabilities of unavailable discrete actions. For instance, to align with regulations in certain countries, two channel operations may be accessible to general cargo while only a dedicated channel is available for electric cargo. In this case, with the operation probability of [0.10, 0.30, 0.60] for [general channel A, general channel B, electrics channel], the masked probability for a general cargo is [0.25, 0.75, 0.00] while the masked probability for an electric cargo is [0.00, 0.00, 1.00]. Accordingly, an electric cargo is only allowed to be processed by the third operation. Incorporating specific government regulations as additional masks on action spaces could provide a natural way to build in relevant constraints. We appreciate you highlighting this potential model enhancement.
>
> **Comment 2. The authors can also look at this complicated problem from a perspective of dynamic mechanism design. The algorithm of the e-commerce platform is mechanism. The platform is searching an optimal mechanism to minimize the cost.**
>
> ***Response.***
>
> Thank you for the insightful suggestion to explore mechanism design perspectives in e-commerce platforms. The current model incorporates the e-commerce platform in the order handling stage, where order processing takes a fixed unit of time and is charged at a predetermined price based on the service level. As you astutely noted, developing dynamic order handling mechanisms could be a fruitful avenue for future work.
>
> **Comment 3. There is a trend along which researchers connect the mean-field game to reinforcement learning. I think that the authors can mention this point in the paper.**
>
> ***Response.***
>
> Thank you for the insightful suggestion to connect our work to mean-field game theory. As you recommended, we have added content to the revised manuscript highlighting how OFCOURSE can model order fulfillment as a mean-field game when agents are homogeneous. Specifically, lines 305-307 now state: "When agents are homogeneous, OFCOURSE also supports modeling order fulfillment as a mean-field game." We appreciate you identifying this relevant approach and advising us to mention it in the paper. We would be grateful for any suggestions you may have regarding specific papers we should cite.
>
> Thank you again for your insightful feedback and suggestions. Please let us know if you have any other thoughts on improving the manuscript and we are happy to update it accordingly. We greatly appreciate you taking the time to thoroughly review our work.

---

### Official Review · Reviewer_D3MQ · 2023-07-26
**The problem is interesting, but the documentation is not clear and the resemblance to real world system is not elaborated.**

**Rating:** 4
**Confidence:** 4

**Strengths:**

The considered problem is interesting and  important. This research of this problem can encompass lots of business scenarios.

**Additional Feedback:**

The problem is interesting, but the reviewer is struggling with associating the variables in Markov games with that in real-world OF systems, which is also not clearly elaborated in section 4.

In addition, the authors used Markov games, but the considered algorithms, HAPPO, are for common payoff games. Thus, Markov game is not necessary relevant in this case.

See Opportunities For Improvement for more details.

**Clarity:**

The writing is in general clear while the reviewer found the structure confusing and some important information is missing, such as how the design of such an OF environment helps solve real world problems.

**Correctness:**

The main contribution of this paper is the development of one order fulfillment environment. The reviewer found it difficult to evaluate its correctness without the domain knowledge on logistics in e-commerce.

The authors will need to take this into account to improve its presentation of the context of the problem.

**Documentation:**

The dataset used for evaluation in task 1 is not released due to the disclosure policy. But the documentation is not clear to the reviewers, e.g. the format of the data, the size, the scenario, and why they selected this scenario.

**Ethics:**

No.

**Limitations:**

No. The authors did not mention these in the paper.

**Opportunities For Improvement:**

* While the overall structure is clear, and the results are also very informative, the reviewer found the description of the order fullfill task is not clear enough.
* This paper mentioned several tasks, including handling, packing and pickup, storage, order consolidation, etc. but I did not see all of them included in the MARL framework, or specifically, how they come into the Fulfill agent described in section 4.5.
* Section 4 includes the description of both tasks and agents, which is confusing to me. The structure could be improved.
* What are the agents in this environment? What are multiple agents? What are they supposed to collaborate on?
* Instead of stating how the OFCOURSE environment is built up, it is very important to also state how this resembles the order fulfillment system in the real world, and how the algorithms developed based on OFCOURSE can be generalized to real world scenarios. For example, what’s the observation, what’s the cost? How are they defined?

**Relation To Prior Work:**

Yes, it is clearly addressed

**Summary And Contributions:**

This paper addresses the problem of efficient order fulfillment which is essential in e-commerce scenarios,  which entails a series of  interdependent online sequential decision-making tasks. Instead of treating these tasks separately, the authors propose a multi-agent reinforcement learning-based method to solve them as a cohesive whole, incorporating tasks like order handling, packing and pickup, storage, order consolidation, and last-mile delivery.
This paper models the integrated problem as a Markov game, and proposes to create an OFCOURSE environment to facilitate the research of efficient solutions for this problem. The authors also designed joint policy that outperforms independent policies in reducing the cost and time.

---

> ### Author Response · Authors · 2023-08-19
> **Response to Reviewer D3MQ (Part 2/2)**
>
> **Comment 4. Instead of stating how the OFCOURSE environment is built up, it is very important to also state how this resembles the order fulfillment system in the real world, and how the algorithms developed based on OFCOURSE can be generalized to real world scenarios. For example, what’s the observation, what’s the cost? How are they defined?**
>
> ***Response.***
>
> We are grateful for your insightful comment. We fully understand the significance of illustrating the real-world relevance of the OFCOURSE environment and how the developed algorithms can be extrapolated to real-world scenarios. In direct response, we have worked to bridge this crucial gap by articulating the translation of order fulfillment elements into the corresponding Markov game framework. As we response to other comments, this detailed mapping process is now in Section 3 of the revised paper. For your convenience, we offer the definitions of observation and cost components as follows:
>
> - Observation: the agent's observation space comprises the aggregated observations of its constituent fulfillment units, with each unit's observations encompassing its containers and operators.
> - Cost: the total fulfillment cost for an order comprises the summed step-wise cost accrued plus any applicable overtime penalty fees.
>
> Please explore Section 3 for a comprehensive elaboration on these definitions and their practical implications. We hope that these clarifications have sufficiently addressed your considerations regarding connections to practical systems.
>
> **Comment 5. The dataset used for evaluation in task 1 is not released due to the disclosure policy. But the documentation is not clear to the reviewers, e.g. the format of the data, the size, the scenario, and why they selected this scenario.**
>
> ***Response.***
>
> Thank you for noting the need to better document our datasets. Please find the dataset in the following addresses.
> - The environments for Task 1 and 2: https://github.com/GitYiheng/ofcourse/blob/main/env/define_exp1_env.py and https://github.com/GitYiheng/ofcourse/blob/main/env/define_exp2_env.py.
> - The examples of data format: https://github.com/GitYiheng/ofcourse/blob/main/docs/act_obs.md.
> - The exact sizes of the observation-action spaces in Task 1 and 2: https://github.com/GitYiheng/ofcourse/blob/main/docs/exp_act_obs.md.
>
> The reasons for selecting Tasks 1 and 2 derive mainly from replicating real-world complexities. Task 1 demonstrates OFCOURSE's capacity to handle mixed physical and virtual orders, as occurs in practice. Meanwhile, Task 2 exemplifies a complex fulfillment task characterized by its extended and diverse fulfillment stages, which is commonly encountered in reality.
>
> **Comment 6. The problem is interesting, but the reviewer is struggling with associating the variables in Markov games with that in real-world OF systems, which is also not clearly elaborated in section 4.**
>
> ***Response.***
>
> We appreciate you point out this problem interesting and feel sorry about the confusion. We think this comment is also related to the mappings from real-world fulfillment concepts to Markov game concepts, like Comment 2,3 and 4. Similarly, please refer to Section 3 of the revised manuscript, where the variables of Markov game, including the action space and observation space have been explained. We hope our revised description can clear your confusion.
>
> **Comment 7. In addition, the authors used Markov games, but the considered algorithms, HAPPO, are for common payoff games. Thus, Markov game is not necessary relevant in this case.**
>
> ***Response.***
>
> Thanks for stating your comments about applying HAPPO to Markov games. We noted that HAPPO was not just for common-payoff games but also commonly applied to Markov games.
> - For example, in page 3 line 6 of [Wen, M., Kuba, J., Lin, R., Zhang, W., Wen, Y., Wang, J., & Yang, Y. (2022). Multi-agent reinforcement learning is a sequence modeling problem. Advances in Neural Information Processing Systems, 35, 16509-16521.], the authors stated that "cooperative MARL problems are often modeled by Markov games" in page 4 section 2.3 line 10 of the same paper, the authors described HAPPO as one applicable algorithm.
> - As another example, in page 15 lines 26-28 of [Fu, W., Yu, C., Xu, Z., Yang, J., & Wu, Y. (2022). Revisiting Some Common Practices in Cooperative Multi-Agent Reinforcement Learning. In International Conference on Machine Learning (pp. 6863-6877). PMLR.], the authors also applied HAPPO to six Markov games.
>
> We hope you find our detailed responses satisfactory. If so, we would be grateful if you would consider adjusting your review rating accordingly. Additionally, please do not hesitate to provide any further suggestions, as we are eager to make improvements based on your insightful feedback.

---

> ### Author Response · Authors · 2023-08-19
> **Response to Reviewer D3MQ (Part 1/2)**
>
> We greatly appreciate you taking the time to thoroughly review our manuscript. Your comments have helped us improve the clarity and presentation of our work. According to you comments, we have revised the paper, and for your convenience, we have marked the revised content in ${\color{blue}blue}$ (please check our revised-version manuscript). Please find our detailed point-by-point responses below:
>
> **Comment 1. While the overall structure is clear, and the results are also very informative, the reviewer found the description of the order fulfillment task is not clear enough.**
>
> ***Response.***
>
> We agree that the previous manuscript could benefit from a clearer, more consolidated explanation of the order fulfillment task. To address this, we have made the following key improvements in the revised version:
>
> - Upfront, we now explicitly state that the core aim of order fulfillment task is to minimize overall fulfillment cost (lines 94-95).
> - We attached a detailed narrative walking through the fulfillment process for an individual order, which illustrates the involved sequential decisions (lines 96-109).
> - Furthermore, we have visually depicted the order fulfillment task in Figure 1 accompanied by a textual description delineated in lines 132-152.
>
> **Comment 2. This paper mentioned several tasks, including handling, packing and pickup, storage, order consolidation, etc. but I did not see all of them included in the MARL framework, or specifically, how they come into the Fulfillment agent described in section 4.5. Section 4 includes the description of both tasks and agents, which is confusing to me. The structure could be improved.**
>
> ***Response.***
>
> We appreciate you highlighting the need to clearly link the fulfillment sub-tasks to the agent framework. To address this, we have made the following improvements in the revised version:
>
> - Before presenting the details of the MARL framework, Section 3 provides an description of the order fulfillment process for a single order, highlighting the various decision-making tasks involved. Following that, we resemble the process and the problems as a Markov game and introduce the necessary concepts and notations to build the MARL framework, including fulfillment unit and agent. We now explicitly state that each sub-task (handling, packing and pickup, storage, order consolidation) is realized as a fulfillment unit, and the concatenation of them defines an agent in the MARL framework detailed in Section 4.5. Please refer to lines 135-138 for the details.
> - After introducing all the components of the MARL framework, we overview the mapping between it and the order fulfillment sub-tasks, please refer to lines 286-293.
>
> **Comment 3. What are the agents in this environment? What are multiple agents? What are they supposed to collaborate on?**
>
> ***Response.***
>
> Thank you for noting that the agent definitions could be clearer. Since this comment is closely related to Comment 2, please refer to that response for an agent is explicitly defined as the concatenation of fulfillment units. Regarding for what multiple agents are and how they collaborate, we have provided additional clarification in the revised version of the manuscript:
>
> - Each fulfillment route is associated with a single fulfillment agent. Considering the presence of numerous fulfillment routes, it consequently introduces multiple fulfillment agents within the system. The crux of collaboration materializes within shared units such as warehouses, where these agents synergize efforts to optimize resource allocation across their respective orders.

---

### Decision · Program_Chairs · 2023-09-22

**Decision:**

Accept (Poster)

**Comment:**

The paper presents a very interesting formulation and benchmark for the order fulfillment problem and present a benchmark environment that is based on the existing and popular OpenAI gym environment. 3 out of 4 reviewers are in favor of accepting the paper, while at the same time providing reasonable suggestions for improving the benchmark to make it more favorable to the RL community as well, which I agree with. The concern from the last reviewer is regarding Markov game formulations, which the authors appear to have addressed with no response. In summary, I am in favor of accepting the paper with suggested revisions for clarity.